# Mapping Tyrosine Kinase Receptor Dimerization to Receptor Expression and Ligand Affinities

**Spencer B. Mamer [1] , Alexandra A. Palasz [1] and P. I. Imoukhuede [2],***

[1] Department of Bioengineering, University of Illinois at Urbana-Champaign, Urbana, IL 61820, USA; smamer2@illinois.edu (S.B.M.); palasz2@illinois.edu (A.A.P.)

[2] Department of Biomedical Engineering, Washington University in St. Louis, St. Louis, MO 63105, USA

* Correspondence: imoukhuede@wustl.edu; Tel.: +1-314-935-7038

**Abstract:** Tyrosine kinase receptor (RTK) ligation and dimerization is a key mechanism for translating external cell stimuli into internal signaling events. This process is critical to several key cell and physiological processes, such as in angiogenesis and embryogenesis, among others. While modulating RTK activation is a promising therapeutic target, RTK signaling axes have been shown to involve complicated interactions between ligands and receptors both within and across different protein families. In angiogenesis, for example, several signaling protein families, including vascular endothelial growth factors and platelet-derived growth factors, exhibit significant cross-family interactions that can influence pathway activation. Computational approaches can provide key insight to detangle these signaling pathways but have been limited by the sparse knowledge of these cross-family interactions. Here, we present a framework for studying known and potential non-canonical interactions. We constructed generalized models of RTK ligation and dimerization for systems of two, three and four receptor types and different degrees of cross-family ligation. Across each model, we developed parameter-space maps that fully determine relative pathway activation for any set of ligand-receptor binding constants, ligand concentrations and receptor concentrations. Therefore, our generalized models serve as a powerful reference tool for predicting not only known ligand: Receptor axes but also how unknown interactions could alter signaling dimerization patterns. Accordingly, it will drive the exploration of cross-family interactions and help guide therapeutic developments across processes like cancer and cardiovascular diseases, which depend on RTK-mediated signaling.

**Keywords:** RTK signaling; dimerization; ligand-receptor kinetics; computational modeling

## 1. Introduction

Tyrosine kinase receptors (RTKs) and their ligands are key to regulating growth, motility and differentiation processes, including: fibroblast growth factors (FGFs), epidermal growth factors (EGFs), vascular endothelial growth factors (VEGFs) and platelet-derived growth factors (PDGFs) [1–6]. RTKs are transmembrane proteins which transduce external signals to internal transduction pathways when an external ligand molecule binds a receptor to induce dimerization [1,7,8]. Different ligands can induce unique receptor conformational changes that are key to allowing downstream signaling activation [9]. Likewise, RTKs can form dimers with identical or different receptor monomers, each activating unique downstream pathways [10]. Therefore, mapping RTK ligation and dimerization patterns across conditions offers a promising means to understanding and manipulating key critical physiological processes.

One critical process governed by RTK ligand:receptor dynamics [11] is angiogenesis, the process of new blood vessel formation from existing vasculature. Angiogenesis plays a critical role

in normal physiological functions as well as in diseases, such as peripheral artery disease and cancers [12,13]. Multiple families of RTKs and their ligands are known to modulate angiogenic behavior, including the vascular endothelial growth factors (VEGFs) and the platelet-derived growth factors (PDGFs) [14–16]. However, VEGF-or PDGF-targeted therapies are not clinically effective for many patients or angiogenesis-dependent diseases [17–19]. As such, there is a need to further understand how VEGF-and PDGF-promoted angiogenesis can be mechanistically controlled to improve the efficacy of current angiogenic treatments. Therefore, quantitatively mapping VEGF and PDGF binding distributions would predict angiogenic responses, allowing higher efficacy angiogenesis therapeutics to be developed.

Two primary challenges impede our ability to map ligand: receptor binding distributions in RTK systems like the VEGF/PDGF signaling axes quantitatively. Firstly, each RTK signaling family involves an array of ligand: receptor interactions between multiple ligands and receptors. The vascular endothelial growth factor (VEGF) family is composed of five ligands, VEGF-A,-B,-C,-D and placental growth factor (PlGF) and three receptor tyrosine kinase monomers, VEGF-R1,-R2, and-R3 [20], that canonically interact thus: VEGFR1 binds VEGF-A and-B; VEGFR2 binds VEGF-A,-C, and -D [21]; and VEGFR3 binds VEGF-C and -D [21,22]. However, the VEGFR monomers can homodimerize, bind the same monomer type, and heterodimerize, bind different monomer types, enabling differential VEGF binding distributions with varying degrees of specificity. VEGFR1-VEGFR2 heterodimers can form through VEGF-A binding, while VEGFR1-VEGFR3 heterodimers can form with VEGF-C and VEGF-D [23,24]. Likewise, the PDGF signaling family consists of five ligands: PDGF-AA,-AB,-BB,-CC, and-DD and two receptors: PDGFRα and-Rβ [25]. PDGF proteins show ligand-receptor and receptor-receptor binding patterns similar to the VEGF family, reviewed in References [25–29]. These specific ligand-receptor and receptor-receptor binding patterns trigger unique downstream signaling, such as the well-established PI3K/Akt and MER-ERK pathways (reviewed in Reference [30]), resulting in different angiogenic responses. Mapping these pathways requires a modeling approach to detangling the effects of each interaction.

The second challenge limiting attempts to fully-characterize RTK dimerization is the prevalence of cross-family interactions. Cross-family or non-canonical interactions—binding of ligands and receptor from different, distinct growth factor families—have been increasingly identified recently in RTK signaling systems. For example, direct interactions between VEGF and PDGF ligands and receptors have recently been identified. Studies have demonstrated that VEGFR2 forms heterodimers with PDGFRβ [16,31] and that VEGF-A can directly bind and signal through both PDGFRs [31]. Additionally, recent work from this lab has demonstrated that several PDGF ligands—PDGF-AA,-AB,-BB, and-CC—can directly bind VEGFR2 [32]. These cross-family interactions could play a significant role in modulating angiogenesis. Therefore, characterizing how cross-family interactions alter dimerization distributions will be key to developing angiogenesis therapeutics. Cross-family ligand: receptor binding and cross-family dimerization patterns introduce a degree of complexity to modeling RTK signaling systems that is rendered especially difficult where the existence of these interactions remains unknown.

Our capacity to model RTK ligand: receptor signaling systems, therefore, is limited by the unknown extent of cross-family binding and sparse experimental kinetics data. This goal to predict key cross-family interactions can be obtained through fundamental models that map the space where important interactions may occur and correlates that space to real ligands and receptors. Experiments are increasingly revealing an expanding set of cross-family interactions—and therefore increasing the number of ligands and receptors within a signaling network [10,31–33].Thus, there is a pressing need for a theoretical foundation that generalizes RTK ligand: receptor binding and dimerization patterns across increasingly complex systems—not just for the VEGF-PDGF axis but for any RTK signaling network.

Here, we present a theoretical framework of ligand-receptor binding, generalized for any RTK family. We apply this framework to show how potential cross-family interactions shift signaling potential. We examine how cross-family interactions affect dimerization patterns across systems of

(1) multiple receptors that form heterodimerization pairs, (2) multiple ligands with variable degrees of cross-receptor binding. Within these systems, we computationally predict how differing kinetic parameter ranges directs receptor dimerization as a proxy for receptor activation. This framework allows us to predict key interaction spaces, thereby directing experimental investigation and therapeutic options for controlling angiogenesis.

## 2. Materials and Methods

### 2.1. Ligand-Induced Dimerization Reaction Scheme

Multiple generalized models were constructed for RTK ligand-induced homo-and heterodimerization in systems of variable numbers of unique ligand and receptor types. Receptor ligation and dimerization were mathematically modeled based upon a common mechanism (Figure 1): (1) model interactions occur within a single compartment that represents (a) the surface of the cell membrane where receptors, receptor complexes and receptor dimers are found and (b) the extracellular space where unbound ligand molecules are present. (2) Ligand-induced dimerization begins with a free ligand (L) binding a single unbound receptor (R) to form a ligand-receptor complex ($C_L$). (3) A ligand-receptor ligand complex can then bind another free receptor monomer to form a receptor dimer (D). The interactions in each model class are represented as a set of ordinary differential equations (ODEs) derived using the law of mass action [34]. (See Supplementary Text 1 for a complete list of model ODEs.) The model equations and parameters are implemented in MATLAB using the SimBiology toolbox. Steady-state and dynamic solutions are computed using the *sundials* ODE solver for 24 simulated hours.

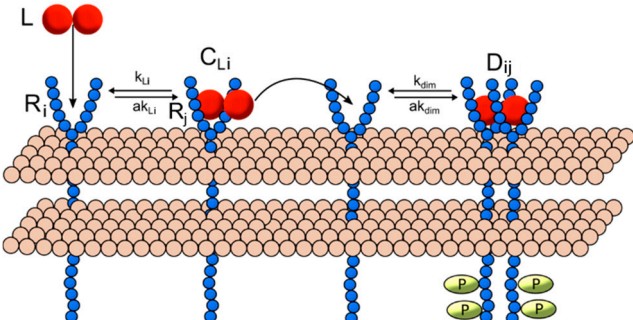

**Figure 1.** Receptor dimerization reaction scheme. Dimerization includes the following reactions for any two receptor monomers with indices **i** and **j**: ligand **L** binds a non-ligated receptor monomer **$R_i$**, with on-rate **$k_{Li}$** and off-rate **$ak_{Li}$**, forming a ligand-receptor complex **$C_{Li}$**. This complex can then dimerize with another un-ligated receptor monomer **$R_j$**, with forward rate **$k_{dim}$** and reverse rate **$ak_{dim}$**, to form the dimer **$D_{ij}$**. Note that **i = j** indicates a homodimer and **i ≠ j** indicates a heterodimer.

### 2.2. Model Parameters

For each model class, we investigated how relative dimerization patterns respond to changes in two physiological parameters known to vary between proteins—relative ligand and receptor concentrations (referred henceforth as $n_i$ and $R_i$, where i indicates the specific ligand or receptor class—and between interactions pairs—the ligand-receptor on-rates ($k_L$) and ligand-receptor off-rates ($ak_L$). In this exploratory model, we model dimerization using a single, static set of dimerization kinetic rate ($k_{dim}/ak_{dim}$) between all homo-and heterodimer pairs, which was adapted from a previous model of EGF/EGFR ligand-receptor interactions. Relative [ligand] and [receptor] concentrations are varied from a baseline concentration of 1 nM and 1000 receptors/cell respectively, based on physiological values found for RTK systems [35]. (See Supplementary Table S1 for a complete list of model parameters and their sources.) We compare the sensitivity in dimerization across binding on-rates, ligand concentrations (in the generalized multi-ligand models) or receptor concentrations

(in the generalized multi-receptor models) for each model by evaluating simulations for different set value ratios (summarized in Supplementary Table S2).

Dimerization is compared between conditions using the relative steady-state dimer, referred to here as the dimer fraction. The dimer fraction is defined Equation (1) as the ratio of the number of steady-state dimer $D_i$ molecules ($n_i$) to the total number of all steady-state dimer molecules in the simulation.

$$f_{D_i} = \frac{n_i}{\sum\limits_{j} n_j} \tag{1}$$

### 2.3. Generalized Multi-Receptor Models

We explored the impact the introduction of additional ligand-binding and heterodimerization-partner receptor types had on dimerization patterns by constructing separate models including two, three and four unique receptor monomers. These generalized receptor models were all constructed under the following restrictions: (1) one unique ligand was available for receptor binding; (2) each unique receptor monomer can bind ligand; (3) a ligand-receptor complex of any receptor type can either form a homodimer with a free monomer of the same type or heterodimer with a monomer of another type. Each additional monomer type increased the number of dimer types produced, for example, three different dimers can result from the generalized two-receptor model: homodimers of each receptor, $D_{11}$ and $D_{22}$ and one heterodimer, $D_{12}$ (Figure 2), while six unique dimers (three homodimers, three heterodimers) can result in the three-receptor model (Figure 3). For each model, simulations were performed for all combinations of different configurations of relative on-binding rates and relative receptor concentrations (summarized in Supplementary Table S2). The model equations governing each model configuration are described in Supplementary Text 1.

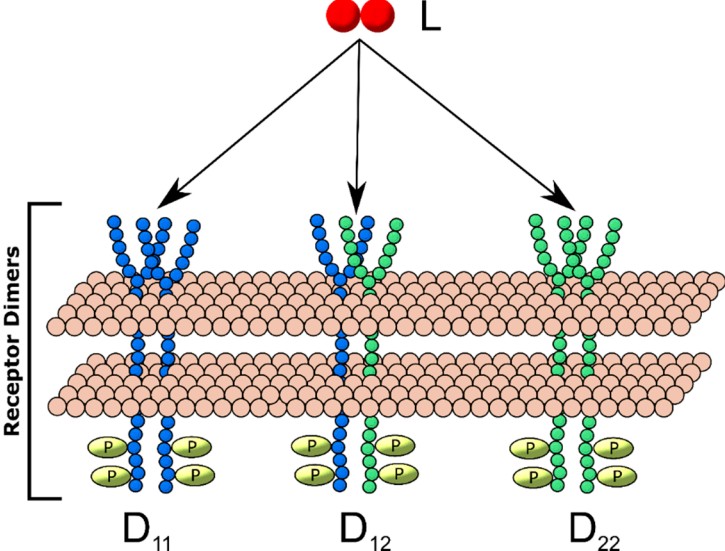

**Figure 2.** Two-receptor, one-ligand (2R1L) dimerization model schematic. Two receptor monomers, $R_1$ (blue) and $R_2$ (red), bind ligand $L$ and form dimers $D_{ij}$, where $i$ and $j$ represent receptor monomers. Thus, $D_{11}$ is a homodimer containing two $R_1$, $D_{12}$ is a heterodimer containing one $R_1$ and one $R_2$ and $D_{22}$ is a homodimer containing two $R_2$.

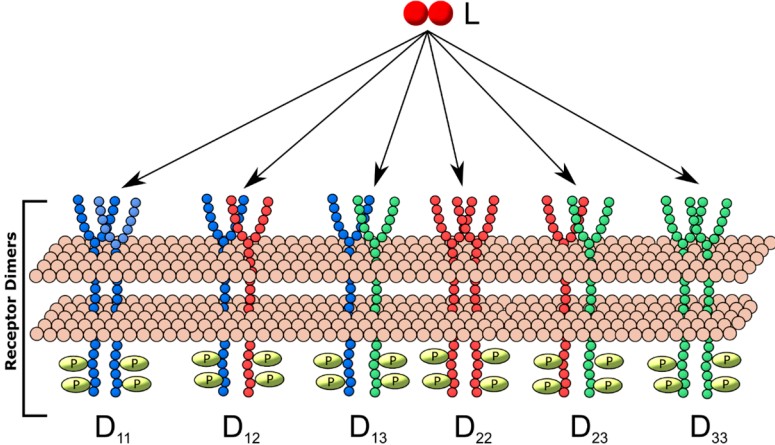

**Figure 3.** Three receptor, one-ligand (3R1L) dimerization model schematic. Three receptor monomers, $R_1$ (blue), $R_2$ (red) and $R_3$ (green), bind ligand $L$ and form dimers $D_{ij}$, where $i$ and $j$ represent receptor monomers. Thus, $D_{11}$ is a homodimer containing two $R_1$, $D_{12}$ is a heterodimer containing one $R_1$, etc.

## 2.4. Generalized Multi-Ligand Models

Cross-family interactions, such as those found between the VEGF and PDGF families, can involve binding of multiple unique ligands to one or more common receptor type. We explored the dimerization patterns resulting from different binding configurations between two ligands and one- and two-receptors: (A) two unique ligands interacting with a single receptor; (B) two receptors that each bind a unique ligand, with no cross-interactions—i.e., Ligand A binds Receptor 1, Ligand B binds Receptor 2; (C) Two receptors and two ligands, with cross-family competition over one receptor—i.e., Ligand A binds Receptors 1 and 2, Ligand B binds Receptor 2; and (C) two ligands both bind each of two receptors—Ligand A binds both $R_1$ and $R_2$ and Ligand B also binds $R_1$ and $R_2$. Each model scheme is summarized in Figure 4; binding rates in red indicate those varied across simulations. We maintained constant and equal receptor concentrations to focus on how different [ligand] configurations and binding rates affected dimerization patterns across different cross-family configurations. We performed steady-state simulations across each [ligand] and $k_A$ configuration.

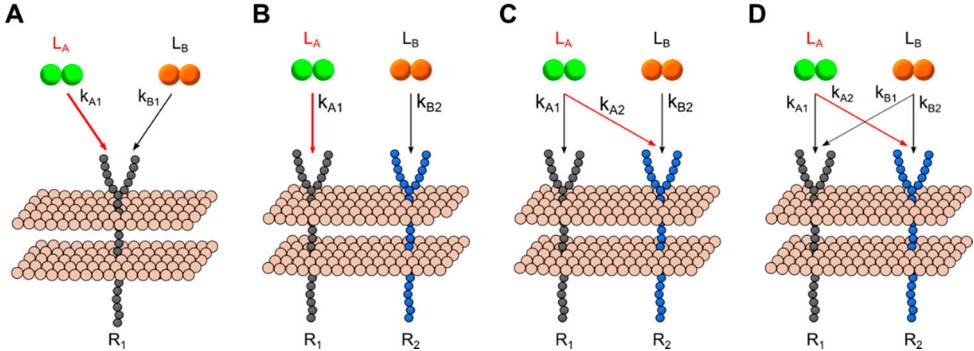

**Figure 4.** Generalized multi-ligand model schemes. (**A**) Two ligand, one receptor (2L1R). (**B**) Two ligand, two receptor, no overlap (2L2R0X). (**C**) Two ligand, two receptor, one overlap (2L2R1X) (**D**) Two ligand, two receptor, two overlaps (2L2R2X).

## 3. Results

We constructed models for several physiologically-relevant ligand: receptor configurations, where one ligand binds two, three and four unique receptor monomer types that can homo-and heterodimerize (representing uni-family binding systems). We explored different cross-family schemes by building models of two ligands and two receptors, either having (a) no overlap in target receptors, (b) one overlapping interaction or (c) two overlapping interactions (Figure 4A–D). For each model,

we performed simulations while varying two major properties: across different ligand-to-ligand relative concentrations or receptor-to-receptor concentrations; and across different relative binding rates between two ligand:receptor interactions. We compiled the simulation results to generate a complete reference table, with which one predict the evolution of any existing physiological system of ligand: receptor interactions and look-up the likely ligand: receptor complexes that would dominate under different conditions (i.e., healthy vs disease serum levels).

### 3.1. A guided Tour Across RTKs: Generalized Two-Receptor Model Demonstrates How Cross-Family Heterodimerization Complicates the Ligand:Receptor Binding Distributions

We constructed a model of the least complex case of cross-family heterodimerization: the two-receptor, one-ligand model (2R1L) (Figure 2). We 'mapped out' the relative homo-and hetero-dimers predicted to form under every combination of initial $k_{L1}/k_{L2}$ and $n_{R1}/n_{R2}$ configurations (Figure 5A). We examine its predictions in detail to illustrate how these generalized RTK models offer insight into how signaling system behavior shifts across different physiological conditions. We first examined 'slices' of this parameter space for specific ratios of ligand: receptor binding affinities (i.e., for constant $k_{L1}/k_{L2}$, the relative kinetic association constants for L:$R_1$ vs L:$R_2$ binding) to observe how physiological changes to [$R_1$] and [$R_2$] (parameters $n_1$ and $n_2$) influence homo- and hetero-dimer formation and can therefore alter dominant signaling pathways activated. When the ligand **L** binds $R_1$ and $R_2$ with equal affinities ($k_{L1} = k_{L2}$), we observe two significant findings: (1) No receptor concentration configuration results in heterodimer ($D_{12}$) domination (Figure 5A); only when [$R_1$] and [$R_2$] are equal (where $n_{R1} = n_{R2}$) do heterodimer concentrations equal to the homodimers $D_{11}$ and $D_{22}$ (~33% for $D_{11}$, $D_{22}$ and $D_{12}$). (2) The homodimer predominantly formed is determined by the highest-concentration monomer. For example, where $n_{R1} > n_{R2}$, its corresponding homodimer $D_{11}$ predominates. Vice-versa, where $n_{R2} > n_{R1}$, $D_{11}$ formation dominates, making up ~90% of all dimers.

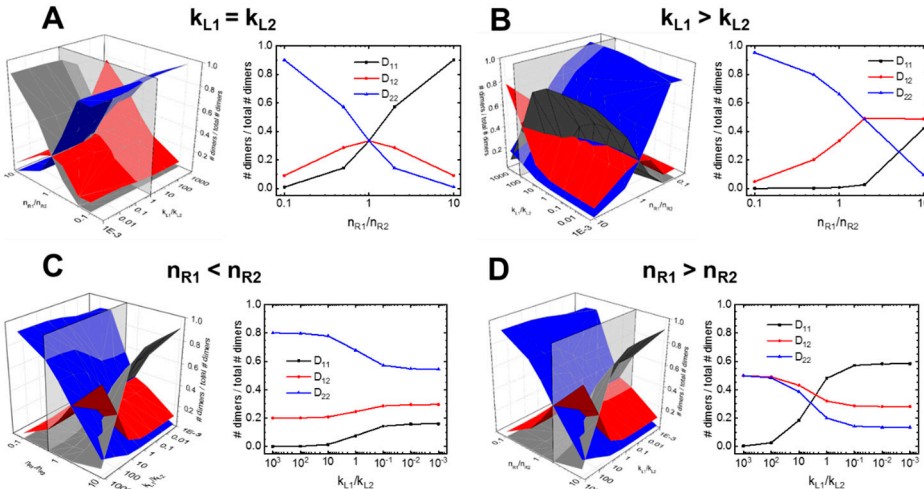

**Figure 5.** Two-receptor, one-ligand model parameter mapping. Predicted steady-state dimer formation across relative affinity- and concentration parameter spaces, depicting changes in fraction dimer expression: across a range of $n_{R1}$ concentrations for (**A**) equal receptor binding affinities and (**B**) where L binds $n_{R1} > n_{R2}$; further, how relative binding rates altered fraction dimer expression where (**C**) $n_{R1} > n_{R2}$ and (**D**) when $n_{R1} < n_{R2}$.

While the predictions made for identical binding rates ($k_{L1} = k_{L2}$) are intuitive, our simulations also explored less intuitive scenarios where a common ligand binds receptor monomers with different affinities—conditions common in RTK binding, such as VEGF-A binding VEGFR1 versus VEGFR2 [32]—and investigated how this altered the relationship between receptor concentrations and homo-and heterodimerization formation patterns. For example, we observe these relationships where $k_{L1} = 100 * k_{L2}$. Here, the stronger L: $R_1$ binding results in diminished $D_{11}$ formation across

all relative receptor configurations (Figure 5B). This behavior appears to be as a result of the effective negative cooperation known in VEGFR and other RTKs; when L predominantly binds free $R_1$, leaving few free $R_1$ molecules available to complete dimerization [36,37], enabling $D_{22}$ dimers to dominate. As $n_{R1}$ increases, $D_{22}$ decreases while $D_{11}$ and $D_{12}$ increase. Notably, once $n_{R1} \geq 2 * n_{R2}$, we see the $D_{12}$ heterodimer become the dominate dimer formed (Figure 5B).

We find that the $n_{R1}/n_{R2}$ concentration ratio modulates the sensitivity of the system to changes in the relative kinetic rate constants. We observe two distinct behaviors or responses to kinetic rate changes that emerge depending upon the relative initial number of receptors. Where $n_{R1} < n_{R2}$, when we vary $k_{L1}/k_{L2}$ from $10^{-3}$ to $10^3$, we do not observe any changes to the order of receptor dominance (Figure 5C). In contrast, we find this behavior changes entirely where $n_{R1} \geq n_{R2}$. Where $n_{R1} = 2 * n_{R2}$, for instance, when we increase $k_{L1}/k_{L2}$, from $10^{-3}$ to $10^3$ the dominant dimer changes from $D_{11}$ to the heterodimeric $D_{12}$, with the transition occurring approximately where $k_{L1} = k_{L2}$ (Figure 5D).

In summary, the 2R1L model serves to illustrate that cross-family interactions introduce complex dynamics relationship between which signaling dimers form and the binding kinetics/cell receptor concentrations particular to the specific RTK family. This model—as will the other generalized models—can serve as a reference table for researchers to predict dimerization profiles. (Further parameter slices can be found in Supplementary Text 1 and a table providing all relative dimer formation versus initial parameter configurations is provided in Supplementary Table S3).

### 3.2. [R]-$k_{L1}$ Relationships Mapped for Higher-Order Models

Ligand: receptor interaction schemes in RTK signaling systems can involve three, four or more unique receptor monomers, each expressed at different relative concentrations and each binding ligands at different affinities. We expanded our model to include systems of three and four receptor monomers and repeated our analysis whereby steady-state dimer levels were predicted for different receptor concentrations and different binding affinities (Supplementary Figure S1). We observe that hetero-and homo-dimerization patterns follow more complicated trends than for the 2-receptor model. Nevertheless, our modeling produced an expanded prediction dataset that enables the use of this tool to quickly identify the predicted dimer distribution for an arbitrary RTK system of multiple receptor monomers (summarized in Supplementary Text 1 and Supplementary Tables S4 and S5).

### 3.3. Dimerization Predictions Expanded for Cross-Family Ligand: Receptor Interaction Systems

In addition to cross-family heterodimerization, there have been increasing occurrences of cross-family ligand interactions that overlap in target receptors. We expanded our model sets to include systems of two ligands with varying degrees of cross-binding to a common set of receptors and analyze how dimerization patterns change for different ligand: receptor binding affinities and ligand concentrations. In the simplest case, when two ligands both bind and compete for one receptor (i.e., in the two ligand, one receptor (2L1R) model), we observe a symmetric relationship across both binding kinetics and ligand concentrations (Figure 6). First, when both ligand concentrations and binding kinetics are equal, we observe equal formation of $L_A$-bound dimers and $L_B$-bound dimers (50% $D_A$ and 50% $D_B$). Within this single receptor system, we observe that ligand-specific dimer formation is equally sensitive to ligand concentrations and ligand: receptor binding kinetics. When $L_A$ is increased relative to $L_B$ or when $k_{A1}$ is larger than $k_{B1}$, $D_A$ formation increases. The presence of cross-family ligand: receptor interactions introduces more complex effects in systems where two ligands each bind a native a receptor while cross-family bind with varying degrees of 'overlap' or number of cross-family interactions to their non-native target receptor. We further analyzed the dimer formation patterns to observe how altering the relative binding affinity each ligand has for the cross-family interaction and the relative ligand concentrations results in different degrees of homo- and hetero-dimer formation (summarized in Figure 7; see Supplementary Text 1 and Supplementary Tables S6–S8 for a detailed breakdown of each parameter space map).

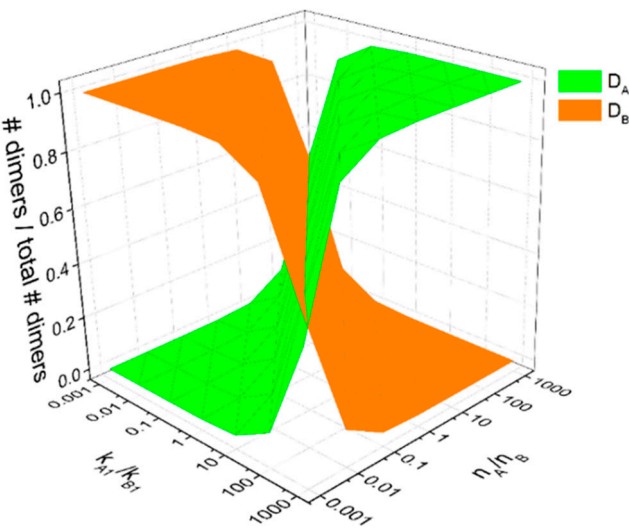

**Figure 6.** Two-ligand, one-receptor model parameter space. The fraction of dimers formed bound with ligand $L_A$ versus bound to ligand $L_B$ shifts as a function of the relative ligand: receptor binding kinetics $k_{A1}/k_{A2}$ and the relative ligand concentrations $[L_A]/[L_B]$.

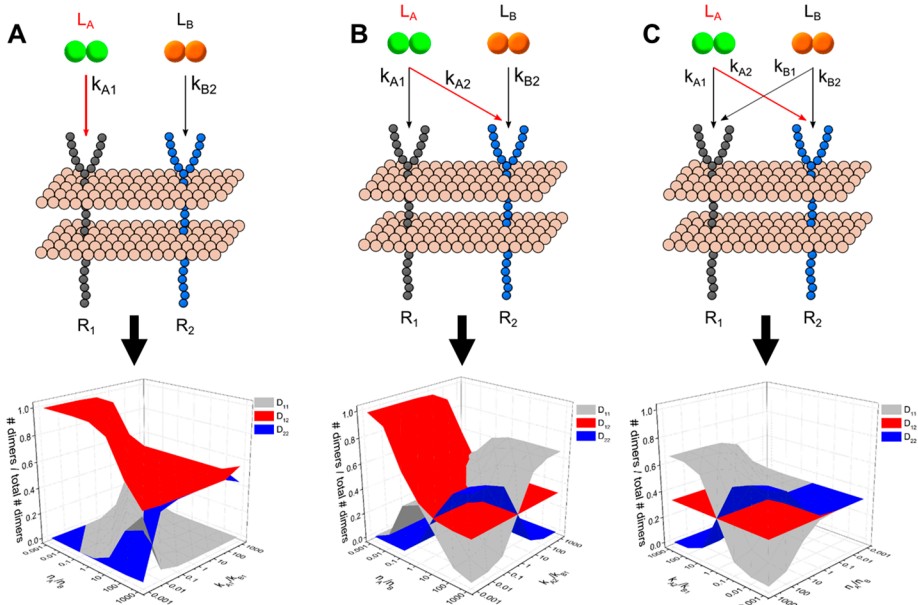

**Figure 7.** Multi-ligand, two-receptor model dimerization patterns across relative affinity-and concentration parameter spaces. (**A**) Two ligand, two receptor, no overlap (2L2R0X). (**B**) Two ligand, two receptor, one overlap (2L2R1X) (**C**) Two ligand, two receptor, two overlaps (2L2R2X).

## 4. Discussion

We constructed models that generalize both simple, single-protein family RTK signaling pathways and progressively more complex systems incorporating multiple signaling families. (1) We modeled generalized systems of two-, three- and four-receptor homo- and heterodimerization patterns. We further modeled two-receptor systems with none, one and two degrees of cross-family interactions. (2) We analyzed the parameter space of receptor concentrations, ligand concentrations, ligand-receptor binding kinetics and degrees of cross-family ligand: receptor binding, mapping out how the system evolves under different configuration. (3) Furthermore, we identified critical points within this parameter space where switches in the dominant pathway occur. The parameter space mappings produced from this modeling provide a novel framework enabling researchers to investigate for investigating the physiological significance of new or suspected cross-family interactions.

### 4.1. Generalized Modeling Predictions Guide Exploration of Specific RTK Systems

Our generalized models of RTK ligation and dimerization provide a novel framework that enables researchers to investigate ligand: receptor systems of new or suspected cross-family interactions. Tyrosine kinase receptor signaling systems like the VEGFs, PDGFs [26,33,38–42] and FGFs [43] involve complex interactions between multiple ligand and receptors. Different degrees of homo-and heterodimerization are observed, so, detangling the influence of individual interactions is rendered difficult by these complexities. Furthermore, cross-family interactions have been observed in fibroblast growth factor/bone morphological protein signaling [44] and between the PDGF/VEGF signaling [29,31,32]. These discoveries highlight the need to predict the impact these existing or newly-discovered non-canonical interactions would have on dimerization and receptor activation patterns.

The results produced from our generalized modeling framework provide an easy-to-use reference for predicting steady-state dimerization patterns for any system of RTK ligand: receptor interactions without needing to construct and evaluate simulations. When studying a specific system, researchers can consult the predictions made for our model for a specified configuration to estimate how the system would evolve. Further, our framework allows researchers to quickly evaluate under what conditions the system would evolve towards a different dominant signaling pathway. For example, we can use these predictions to identify the $[L_1]/[L_2]$ conditions where homodimers versus heterodimers dominate for a particular ligand: receptor RTK system. Likewise, we can consult the model predicts to determine the relative binding affinity a suspected cross-family ligand: receptor interaction would be required to alter the dominant signaling dimers formed. Together, our generalized modeling of dimerization and ligation provide a framework for studying cross-family and inter-family interactions.

### 4.2. Generalized Models Predict Dimerization Patterns Observed Across RTK Systems

Our results expand upon earlier growth factor family-specific models that explored how homo-and heterodimerization patterns. Homo-and heterodimerization have been explored previously for several RTKs, including: epithelial growth factor receptor [45–49] (EGFR) and vascular endothelial growth factor receptors [50] (VEGFRs). These previous models studied how dimerization depended upon receptor concentrations within the scope of the specific receptor tyrosine kinase family. Hendriks et al., for instance, characterized how homo-and heterodimerization patterns between EGFR and HER2 shifted with changes to either receptor concentrations, in a family-specific version of our two-receptor, one-ligand (2R1L) model. Their model observed a shift from predominantly EGFR-homodimers to HER2 homo-and hetero-dimers as [HER2] increased above [EGFR]. We observe the same result within our generalized models, where $n_{R1} >> n_{R2}$ in the 2R1L model (Figure 5A) [46]. In their VEGFR dimerization model, Mac Gabhann demonstrated how VEGFR1 and VEGFR2 homo-and heterodimerization formation changed across different initial ligand and receptor concentrations. Our generalized 2R1L model makes predictions that align with their results, demonstrating that increasing [VEGFR1]/[VEGFR2] from 1:1 to 10:1 would result in VEGFR1 becoming the pre-dominant dimer formed [50].

Both models demonstrated the predictive power of dimerization models for their respective signaling family. However, their results could not be generalized to predict dimerization for any RTK ligand:receptor system. They were instead limited to the kinetics and interaction patterns of the specific RTK family being explored—i.e., using binding patterns and kinetics specifics known for EGF: EGFR/HER2 and VEGF-A: VEGFRs. Our generalized modeling approach—by instead exploring abstractions of different RTK configurations–can predict dimerization patterns for any system. Therefore, they can provide insight into any RTK system of interest by simply plugging in the relevant kinetic and concentration measurements.

### 4.3. Generalized Models Improve Model Reusability

Generalizing systems increases model modularity—and thus, the ability of researchers to mix-and-match or recombine—by reducing them to discrete units stripped of system-specific mechanistic details. The use of modeling has become a key approach towards studying biological and biochemical systems [51,52]. Through computational systems biology approaches, models have been constructed to study a wide-array of biochemical processes such as VEGF signaling [32,53–57]. Recent advances in standardizing model formats have offered the promise of integrating existing models, including the Systems Biology Markup Language (SBML) [58] and CellML [59]. Nevertheless, several hurdles remain to effective interoperability between models of related biological systems, such as biological network data for biological pathways, as well as modeling of systems at different scales [60,61].

One major hurdle is the highly-specific, model-specific definitions of different biological networks [61]. For example, individual computational models of systems often employ model-specific schemes for naming molecules types—for example, ligands versus receptors—or molecules in different states—such as phosphorylation status. Additionally, different biological models can depict interaction networks with different degrees of detail. Together, these differences in detail leave few common interfaces to link models. Semantic annotation approaches offer promising solutions to this impediment but such solutions increase the complexity of constructing models of even simple systems [60,62,63]. In contrast, constructing models that generalize systems—such as RTK ligation and dimerization—biological models are reduced to their basic components—i.e., ligands, receptors and dimers. These reduced models generalize the interaction scheme, enabling them to serve as self-sufficient 'modules' to be combined at ease. For example, the different ligand: receptor interaction pattern models we constructed can serve as 'dimerization modules' to be linked with downstream signaling cascade modules. Therefore, generalized models provide a simpler approach for constructing module representations of biological systems that can be combined with ease.

### 4.4. Generalized Model Enables Exploration of Cross-Family Interactions in Human Disease

Our generalized framework for studying cross-family interactions can be applied to explore how cross-family interactions alter signaling activation in processes important in human disease. For example, angiogenesis is regulated by a complicated signaling network involving both cross-family dimerization and cross-family ligand: receptor interactions: the pro-angiogenic vascular endothelial growth factor receptor 2 (VEGFR2) hetero-dimerizes with a platelet-derived growth factor receptor (PDGFRβ) [64] and recent work by this lab has shown several PDGF ligands (PDGF-AA,-AB,-BB, and–CC) bind VEGFR2 [32]. Fully-characterizing angiogenesis requires untangling the influence each VEGF-PDGF cross-family interaction has on VEGFR2 signaling. These models provide a framework to achieve this aim. By using these model predictions, we can determine which cross-family interactions significantly impact the dimerization and activation of VEGFR receptors equipped with parameters known experimentally, such as the ligand: receptor binding affinities [32,65], receptor concentrations [66,67] and ligand concentrations [32].

### 4.5. Simplifying Models Enables Larger Picture Insights of Complex Biological Systems

Our generalized modeling of RTK ligand: receptor interactions furthers recent demonstrations that simple, more generalized models can be powerful tools to understanding biological networks. Detailed and complex models—like the recent model of VEGFR1 signaling [54]—are powerful tools for characterizing signal transduction networks. Such model construction, however, requires detailed knowledge of (a) the biological interaction network and (b) kinetics and concentration measurements for each component [61]. Early and recent work, however, has demonstrated the insights provided by smaller, simpler models that represent the more complex system. Models of EGFR: HER2 ligation and dimerization provided key insights as to how internalization dynamics

affected dimer formation patterns, which served as proxies for predicting downstream signaling activation [45,46]. Likewise, simple mechanistic models VEGFR ligation demonstrated how PlGF concentrations affected VEGF-A:VEGFR ligation rates and that VEGFR1/VEGFR2 concentrations determined homo- and heterodimerization patterns [50]. Recently, a compact ligand: receptor binding model suggested a key compensatory role for cross-family PDGF:VEGFR2 interactions in anti-VEGF-A drug resistance [32]. These focused, compact models provided key insight into the larger output of the signaling systems.

Our generalized models expand on the predictive power of these earlier simple models. Rules-based modeling approaches have also been employed recently to simplify modeling of interaction networks. Under this scheme, biological interaction networks are generalized by local 'rules' that describe general interactions available for a specific molecule (e.g., ligand-binding, dimerization, phosphorylation or signal transduction activation) without providing explicit chemical reaction schemes for every combination of molecular state [68,69]. Models defined using a rules-based approach are 'simplified' by removing the need to define vast arrays of chemical reactions to describe specific biological processes [63,70]. Together, earlier compact mechanistic models and the newer approaches to simplify model definitions have enabled key insights. Our generalized approach to modeling mirrors these approaches by reducing multiple systems to their common interaction mechanism and exploring how different variations would alter their evolution. Therefore, our generalized models further demonstrate the insights that can be gained by reducing the complexity of biological models.

## 5. Conclusions

RTKs including the VEGFRs, FGFRs, PDGFRs and EGFRs mediate signaling crucial in regulating physiological processes critical to human health. We here presented a generalized framework for studying any arbitrary RTK network. The generalized models we developed can represent and simulate ligation and dimerization for any RTK networks, knowing their kinetics and concentrations. Critically, we provide a framework for exploring how different cross-family interaction schemes alter receptor dimerization patterns. As additional cross-family interactions are discovered, the model predictions will be crucial for determining how these interactions alter receptor activation within these processes. Furthermore, we can further study which cross-family interaction parameters provide a targetable 'dial' that allows for manipulating the process towards pro- or anti-angiogenic therapeutic effects. Researchers can further to determine adjustments needed to adapting therapeutics to the different ligand and receptor concentration conditions found in different angiogenesis-related conditions, such as cancers, cardiovascular diseases and diabetes [41]. By applying our generalized modeling approach to specific conditions, our results enable exploring cross-family interactions across different biological processes critical to human health.

**Supplementary Materials:** The following are available online at http://www.mdpi.com/2227-9717/7/5/288/s1, Supplementary Figure S1. Comparison of dimerization regimes at higher order; Figure S2. One receptor model. Compares fractional dimers activated by ligand A with dimers activated by ligand B across different ligand concentrations while varying the ratio of the binding constants; Figure S2. One receptor model. Compares fractional dimers activated by ligand A with dimers activated by ligand B across different ligand concentrations while varying the ratio of the binding constants; Figure S3. Two receptor model with no overlap. Compares the fractional dimers of homodimers made with receptor 1 and receptor 2, as well as heterodimers made with receptors 1–2 across different ligand concentrations while varying the ratios between the binding rates; Figure S4. Two receptor model with one overlap. Compares the fractional dimers of homodimers made with receptor 1 and receptor 2, as well as heterodimers made with receptors 1-2 across different ligand concentrations while varying the ratios between the binding rates; Figure S5. Two receptor model with two overlaps. Compares the fractional dimers of homodimers made with receptor 1 and receptor 2, as well as heterodimers made with receptors 1-2 across different ligand concentrations while varying the ratios between the binding rates; Supplementary Text 1: Model Equations; Supplementary Table S1: Generalized two, three, and four receptor models default parameter values applied; Supplementary Table S2: Generalized two, three, and four receptor model parameter variation values applied; Table S3. Two-receptor, one-ligand (2R1L) model predictions.

**Author Contributions:** P.I.I., S.B.M., and A.A.P. conceived of the studies. S.B.M. and A.A.P. designed the computational models, performed the simulations, and analyzed the results. S.B.M., A.A.P., and P.I.I. drafted the manuscript and contributed to its revision. All authors have read and approved the final manuscript.

**Funding:** This work was supported by American Heart Association (Grant #16SDG26940002), the American Cancer Society, Illinois Division (Grant #282802), the National Science Foundation (NSF Grant #1743333, NSF #1743334, NSF #1640783, NSF CAREER Award #1653925, NSF CBET #1512598 and NSF BPE #1648454) and the National Science Foundation Cellular and Molecular Mechanics and BioNanotechnology Integrative Graduate Education and Research Traineeship.

**Conflicts of Interest:** The authors declare no conflict of interest. The funders had no role in the design of the study; in the collection, analyses or interpretation of data; in the writing of the manuscript or in the decision to publish the results.

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
