# Peer review of "Mapping Tyrosine Kinase Receptor Dimerization to Receptor Expression and Ligand Affinities"

_processes, doi:10.3390/pr7050288_

Round 1

Reviewer 1 Report

The manuscript “Mapping RTK Dimerization to Receptor Expression and Ligand Affinities” focuses on computational approaches on RTK ligation and dimerization for multiple systems of receptor types, and different degrees of cross-family ligation. The problematic is generally well approached and the the generalized models of RTK ligation and dimerization here described can be fruitful used to investigate ligand:receptor binding, receptor dimerization, and finally cross-family interactions.

A limitation of the work is the lack of cross-validation of the model(s) here described using laboratory experimental data. A significant improvement to the manuscript could be given by the use of this specific class of techniques where empirical data are randomly split into training and testing subsets. In this way the model can be trained (do regression over) using the training set only, and then it is possible to verify how well the trained model (i.e. model plus parameters) does on predicting values for the testing set. Therefore the robustness of the proposed model(s) here described can be definitely and unequivocally tested.

Author Response

Reviewer #1

The manuscript “Mapping RTK Dimerization to Receptor Expression and Ligand Affinities” focuses on computational approaches on RTK ligation and dimerization for multiple systems of receptor types, and different degrees of cross-family ligation. The problematic is generally well approached and the generalized models of RTK ligation and dimerization here described can be fruitful used to investigate ligand:receptor binding, receptor dimerization, and finally cross-family interactions.

Reviewer #1 Comment: A limitation of the work is the lack of cross-validation of the model(s) here described using laboratory experimental data. A significant improvement to the manuscript could be given by the use of this specific class of techniques where empirical data are randomly split into training and testing subsets. In this way the model can be trained (do regression over) using the training set only, and then it is possible to verify how well the trained model (i.e. model plus parameters) does on predicting values for the testing set. Therefore the robustness of the proposed model(s) here described can be definitely and unequivocally tested.

We thank the reviewer for this suggestion. We agree with the reviewer that one way this work could be enhanced would be for it to be applied to specific systems with experimental data and validated. We think this suggestion, though, is outside the scope of what this manuscript is discussing. We provide here a general tool and framework based on well-defined mechanisms for studying specific RTK systems, allowing researchers a reference for the ligation patterns predicted. Future work, however, will apply these framework models to specific RTK systems, such as in angiogenesis. In those applications, the choice of parameters will be critical for the accuracy of the models, and those future studies should be trained off experimental data and undergo global sensitivity analyses to confirm they are robust against the parameter values.

We do compare our general model predictions with those made by specific RTK computational models developed in the past. We agree that we did not emphasize clearly how our models aligned with other predictions. We have updated the Discussion to speak more explicitly about where our model predictions align with existing models that have already been validated experimentally. The Discussion now reads (Page 9, Lines 318-331):

Homo- and heterodimerization have been explored previously for several RTKs, including: epithelial growth factor receptor [45–49] (EGFR) and vascular endothelial growth factor receptors [50] (VEGFRs). These previous models studied how dimerization depended upon receptor concentrations within the scope of the specific receptor tyrosine kinase family. Hendriks et al., for instance, characterized how homo-and heterodimerization patterns between EGFR and HER2 shifted with changes to either receptor concentrations, in a family-specific version of our two-receptor, one-ligand (2R1L) model. Their model observed a shift from predominantly EGFR-homodimers to HER2 homo- and hetero-dimers as [HER2] increased above [EGFR]. We observe the same result within our generalized models, where nR1 >> nR2 in the 2R1L model (Figure 5A) [46]. In their VEGFR dimerization model, Mac Gabhann demonstrated how VEGFR1 and VEGFR2 homo- and heterodimerization formation changed across different initial ligand and receptor concentrations. Our generalized 2R1L model makes predictions that align with their results, demonstrating that increasing [VEGFR1]/[VEGFR2] from 1:1 to 10:1 would result in VEGFR1 becoming the pre-dominant dimer formed [50].

Reviewer 2 Report

The study by Mamer et al., entitled “Mapping RTK Dimerization to Receptor Expression and Ligand Affinities” proposes to construct a receptor/ligand molecular interaction model based on growth factor receptor ligands. It is therefore the modeled pharmacological study of the RTKs with their ligands, in several situations during which the binding of the ligand favors the dimerization of the receptor; the interaction of the ligand with several receptors, homo- or heterodimeric is described; the interaction of several ligands with one or more receptors is also described. This study provides a comprehensive template of pharmacological situations described in terms of affinity and reaction kinetics to predict, according to L/R stoichiometry, the main features of the L-R interaction. This study is rational, very interesting and is a good reference for in vitro pharmacological studies of RTK.

L49 : “As such, there is a further need to further understand”: avoid repetition

Figure 1: Rj does not appear in the figure, but is mentioned in the caption.

Figures 2 and 3: Please indicate R1 (blue), R2 (green), R3 (red) in the figure or as a legend.

Figure 3: two different red colors are used and I don’t kow why (one is pink and looks as if the monomer is behind, whereas the other red appears at the front). The same does not apply for Fig. 2.

L206 : with equal affinities?

Author Response

Reviewer #2

The study by Mamer et al., entitled “Mapping RTK Dimerization to Receptor Expression and Ligand Affinities” proposes to construct a receptor/ligand molecular interaction model based on growth factor receptor ligands. It is therefore the modeled pharmacological study of the RTKs with their ligands, in several situations during which the binding of the ligand favors the dimerization of the receptor; the interaction of the ligand with several receptors, homo- or heterodimeric is described; the interaction of several ligands with one or more receptors is also described. This study provides a comprehensive template of pharmacological situations described in terms of affinity and reaction kinetics to predict, according to L/R stoichiometry, the main features of the L-R interaction. This study is rational, very interesting and is a good reference for in vitro pharmacological studies of RTK.

Reviewer #2 Comment: L49: “As such, there is a further need to further understand”: avoid repetition

We agree with the reviewer that this line is written redundantly. We have revised this statement to be clear, and less repetitive. The revised lines now read (Page 2, Lines 49-51):

As such, there is a need to further understand how VEGF- and PDGF-promoted angiogenesis can be mechanistically controlled to improve the efficacy of current angiogenic treatments.

Reviewer #2 Comment: Figure 1: Rj does not appear in the figure but is mentioned in the caption.

We thank the reviewer for identifying this discrepancy between Figure 1 and its caption. We have updated Figure 1 to include the Rj label as originally intended. Figure 1 has been revised to appear as:

Reviewer #2 Comment: Figures 2 and 3: Please indicate R1 (blue), R2 (green), R3 (red) in the figure or as a legend.

We thank the reviewer for this suggestion, which improves the clarity of the model schematics. We have updated Figure 2 and Figure 3 to indicate which color indicates which receptor monomer in the figure caption. In addition, we have updated the colors to be consistent between Figure 2 and Figure 3, such that R1 and R2 are blue and red between each.

The revised Figure 2 and Figure 3 now appear as:

Figure 2. Two-receptor, one-ligand (2R1L) dimerization model schematic. Two receptor monomers, R1 (blue) and R2 (red), bind ligand L and form dimers Dij, where i and j represent receptor monomers. Thus, D11 is a homodimer containing two R1, D12 is a heterodimer containing one R1 and one R2, and D22 is a homodimer containing two R2.

Figure 3. Three receptor, one-ligand (3R1L) dimerization model schematic. Three receptor monomers, R1 (blue), R2 (red), and R3 (green), bind ligand L and form dimers Dij, where i and j represent receptor monomers. Thus, D11 is a homodimer containing two R1, D12 is a heterodimer containing one R1, etc.

Reviewer #2 Comment: Figure 3: two different red colors are used, and I don’t know why (one is pink and looks as if the monomer is behind, whereas the other red appears at the front). The same does not apply for Fig. 2.

We agree with the reviewer that the coloring in Figure 3 is confusing. The different shades of red did not convey any additional meaning, and we have updated the figure to use the same coloring for each monomer, regardless of it’s in-front or behind in the dimer. Additionally, we have updated the color scheme in Figure 3 to be consistent with Figure 2, such that R1 is blue in both, R2 is red in both. The updated Figure 3 now appears as:

Reviewer #2 Comment: L206: with equal affinities?

We thank the reviewer for pointing out this typographical error. We did indeed mean to say “with equal affinities” as opposed to “and equal affinities”. We have revised the text to reflect this change, and now reads (Page 6, Line 205-206):

When the ligand L binds R1 and R2 with equal affinities (kL1 = kL2), we observe two significant findings…
